# The Effect of Destination Brand Identity on Tourism Experience: The Case of the Pier-2 Art Center in Taiwan

**Chien-Ting Chiang and Ying-Chieh Chen \***

Department of Leisure and Sport Management, Cheng Shin University, Kaohsiung City 833301, Taiwan
\* Correspondence: deakin2007@gmail.com

**Abstract:** This paper examines the tourism destination brand identity and brand experiences which can influence visitors' intention to recommend. The study of the importance of destination brand identity and brand experiences in the context of Taiwan has shaped some important insights with the potential to enhance the attractiveness of cultural and creative sectors. In this study, this paper explores perceptions of destination brand identity (tourism brand perception and tourism brand self-concept) and brand experiences. The analysis draws upon data collected at the Pier-2 Art Center in Taiwan in 2019, using a self-administered questionnaire survey. Both confirmatory factor analysis (CFA) and structural equation modeling (SEM) were applied. It has been found that the role of various constructs of a brand perception and a brand self-concept of the tourism brand identity during a visit to cultural and creative parks is on top of the list of concerns associated with visitors' brand experience. An examination of the research comments concluded that the cultural and creative tourism sector about consumer demands and update the development of appropriate marketing strategies, thereby providing visitors to experience the brand characteristics within the creative arts sector.

**Keywords:** tourism destination brand identity; brand experience; intention to recommend; confirmatory factor analysis; structural equation modeling (SEM); arts and culture festivals; tourism

## 1. Introduction

According to the World Travel & Tourism Council (2022) statistics, the tourism industry contributed 10.3% of global GDP 2019 [1]. WTTC states travel & tourism as one of world's largest economic sectors, creating one in ten jobs worldwide in 2019. Also, one in five new jobs were created in the last five years by the travel & tourism industry. Tourism has increased particularly promptly with Asia and has provided opportunities to boost social relationships through relaxation and recreations. Many cities have now developed festivals that have a distinct local feel, notably by incorporating a culinary component. Such approaches have expanded the range of appealing destinations and can help to create an improved Taiwan's tourism market positioning, as well as delivering economic benefits, creating jobs, and extending the tourist season thereby enhancing viability. Festivals and related activities bring large numbers of tourists into a city, driving changes in both the tourists and the city environment [2–4]. Festivals have a significant impact on the economic level, the political level, the level of social interaction, and the culture [4].

According to the United Nations Educational, Scientific and Cultural Organization, UNESCO) "Cultural Industries" mostly refers to the combination of creation, production and commercial. The practice of cultural and creative industry can be as goods or services. Arts and culture tourism describes visits to museums, galleries, arts festivals, buildings, historic sites, literature, art performances, and folk culture. A survey by the Directorate General of Budget, Accounting and Indicators (cited in Chang, Hsiao, Wu 2003, p. 55) reported that the most preferred lifestyle for the Taiwanese involves the maintenance of good health (59%), followed by enjoying family time (26.7%) and having a wealthy life

(26%) [5]. In 2008, Taiwanese authorities developed "National Development Key Plan" to support cultural and creative industry development to create inventive products with local cultural features. The majority of the policy was to enhance people's cultural interactions by combining design creativity with artistic production. Taiwan's creative parks are mostly arts and design sites. Many written explanations indicated that the focus on cultural and creative industry feature turning exhibitions in various creative fields like photography, illustration, film, animation, product design, and furniture design, perhaps even music and performance art.

That festivals not only stimulated the demand for tourism, but also for off-season tourism, by bridging the gap between domestic tourists, and providing a key demand alternative [6]. The series of steps in the cognitive cultural process that underlies the process of cultural change that has come to be known as 'city branding' [7]. Many scholars suggest that organizing local events is the most important factor in increasing sightseeing and attracting visitors, and to create a set of distinctive and favorable associations in consumer memory as a way to differentiate and compete with different destinations [6,8].

Many scholars have focused on sightseeing that involved festivals or competitions [8–11]. Despite these limitations, surprisingly, there has been far less research on the concept of a tourism destination brand identity of arts and cultural festivals' tourists, of which involves an interaction between a tourist and arts and cultural festival, based on the particularity of the creative parks. Responding to market growth, tourism brand identity is for tourists who agree with the value of the festival's brand as expressed through its ideas, beliefs, and marketing strategies, as this includes a specific image name, symbols, patterns, and text characteristics. Scholars identify a major role of cultural and historical heritage of a city in relating to visitors' preference of their holiday decisions. Proving great major resource of a destination to attract tourists would be better meeting their expectaions and needs for the local tourism authority [5]. According to survey findings, communication with the destination via an unforgettable travel experience and enhance the emotional connection between the tourist and the destination of the festival [12].

The current research focuses on the Kaohsiung Arts Spring Festival (KSAF), a notable initiative which projects an international and innovative image for Taiwan's second city. In the five years since its establishment, KSAF has become the most eye-catching festival for conveying Kaohsiung's brand). KSAF is a successful local arts and cultural festival in the Asian context.

Furthermore, a better understanding of art and creative tourism brand identity will be useful in the development of destination marketing because of its effects on both the brand experience and the recommendation of visitors. Hence, to address this gap in the literature, the current study examines the critical role of the tourism brand identity of arts and culture festivals on tourist intention to recommend via a positive brand experience. The link with the tourist experience is critical. In a globalized society where 'different' and 'unique' mean everything, brands are having a moment of glory [13]. The brands which are most distinguishable from alternatives are those that offer an association with the self-image of participants. A review of previous studies [14,15] has shown that branding can support product diversity and hence build up destination competitiveness. However, empirical data suggest that the same plan or urban policy can be interpreted selectively during the implementation phases of different government agencies [16]. So Zarantonello & Schmitthave pointed out that through the process of tourism consumption, brand experience arouses inner and behavioral responses of tourists [9]. The preceding section suggests that festivals are directly associated with destination brands and that tourist experiences are essential to this connection. For a theoretical perspective, the literature has shown that two key different determinants of visitors' antecedents, namely tourism brand identity and brand experiences which are relative to interactive responses. This research should provide some insights into the potential future directions for Taiwan's art and creative tourism sector. Thus, when tourists participate in destination's arts and culture festivals, the festival's name, symbols, graphics, and traits with specific meanings can create an un-

forgettable travel experience, enhancing the emotional connection between the tourist and the destination. As a result of the above discussion, the following hypotheses are presented:

**H1.** *Tourism destination brand identity positively influences brand experience.*

**H2.** *Tourism destination brand identity positively influences intention to recommend.*

**H3.** *Brand experience positively influences intention to recommend.*

## 2. Conceptual Framework and Literature Review

### 2.1. Destination Brand Identity of Art and Creative Tourism

Branding is an important issue that has attracted scholarly attention [17–21]. Though the concept of tourism destination brand identity is based on organizational identity theory, it is related to the self-brand connection [22]. Brand identity was first proposed by Kapferer's brand identity prism concept, the exterior and the interior identity of the brand would connect the consumers' self-image, and the brand's exterior image through an interactive relationship and culture. Brand identity is a combination of position and characteristics, making the product or the service outline a unique attractiveness in the consumers' emotions [23].

While importance of destination brand value has been proposed in the tourism behavior literature, with the both brand value and love marks are found to affect relationship strength among tourists [24]. Brand identity is the most significant and greater part of an organization's assets. It is a tool that guides the brand and gives it inspiration [25]. Furthermore, many scholars have proposed to develop city/destination branding. The strategic development should focus on brand identity and brand equity [21,26–28].

If we understand the perceptions of residents and tourists, knowing that brands are increasingly recognized by local residents will help create an image that attracts tourists to the destination [29]. Zhou et al. indicated that brand identity as a measure of how tourists perceive and recognize brands is a way to assess tourists' perceptions and brand self-concept [30]. In addition, Escalasalso pointed out that consumer brand identity is the degree to which the brand can represent self-concept [22]. Aaker argues brand identity is that makes consumers select to buy products or experience services based on their perception of the brand [14]. Therefore, brands should be able to experience, the feelings and experiences can create a self-consumer mind, with unforgettable value. This paper will evaluate the relationship between tourists' perceptions and brands self-concept as measures of tourism brand identity by the perspective of Zhou et al. [30].

Riza, Doratli & Fasli indicated that using iconic buildings to build city image and brand identity suggests that citizen welfare and visitor satisfaction are strongly influenced by city or local image [31]. Travelers can also visit the attractions and engage in entertainment for more than 24 h so they need accommodation. Therefore, tourist destinations should have suitable amenities [32]. Therefore, according to this study, we can find that tourism destination brand identity has a positive and significant impact on the intention to recommend behavior. The higher the brand identity of tourists, the higher the intention to recommend [30]. Also, That brand identity are the most significant predictors of behavioural intentions of tourists, when users are satisfied with a brand, their identity of the brand enhances their positive perception of the brand and considers them to be a certain degree of community [33]. Therefore, tourism destination brand identity develops when the value of the brand and the sales strategies are identified and trusted by tourists [34]. Overall, Tourism destination brand identity is a complex amalgam of aspect, position, image, idea, name, and the comprehension of tourists. Tourism destination brand identity has been defined as an individual incorporating a brand into their self-concept to some degree [35]. This suggests that the research which combines the marketing literature (focusing mostly on other brand concept) with the tourism literature is still in its initial stages [26].

Accordingly, within Cai's proposed with destination branding the marketing function's role is emphasized, in a tourism destination's brand identity progress special consideration should be given to examining the specific characteristics of the brand as an organization, which should statement the topic of destination culture, its regional people [36] and their relationship. When tourists are satisfied with the brand, their identification with the brand enhances their positive impression, and they will then form a social group with it on some level [37]. Thus, when tourists participate in destination's arts and culture festivals, the festival's name, symbols, graphics, and traits with specific meanings can create an unforgettable travel experience, enhancing the emotional connection between the tourist and the destination. As a result of the above discussion, the following hypothesis is presented:

**H4.** *Tourism destination brand identity certainly influences intention to recommend.*

### 2.2. Brand Experience and Intention to Recommend

Marketing researchers have concluded that tourists' brand experiences are significant determinants of satisfaction, decision making and future behaviors [38]. Brakus, Schmitt and Zarantonellodefined that brand experience as the stimulation can evoke the feelings of sensitively, emotionally, cognitively, and behaviorally [39]. Brand design, cognitive effects, packaging, communication, and environment are all part of the brand. These stimuli mean that brands are subjective and induce the inner and behavioral responses of consumers [39,40]. Barnes, Mattsson and Srensencontended that the brand experience concept had four aspects: sensory, emotional, cognitive, and behavioral [11]. This conception of brand experience includes products, shopping, service, and consumption.

Tourists can generate brand loyalty through building a decent brand image and experience [40]. This builds tourist experience of the destination, promotes satisfaction and loyalty, increases a brand relationship, and stimulates value [40,41]. Brand experience has been defined as highlighting the experiencing value of tourists, with the design, packaging, communication, and environment in the arts and cultural festivals bringing to mind their subjective responses and thereby producing emotion and/or attachment to the destination [11,39]. In summary, the brand experience concept to understand the meaning of brand trust and loyalty and to develop a brand experience scale [39]. Many scholars have responded to the importance of brand loyalty by highlighting in an effort to keep customers, while modifying their image and products to meet the perceived wishes of a growing tourism market [38]. Consequently, brand experience plays an important role in the tourism industry and should receive greater attention, since it is directly related to the usage of the resources, and their satisfaction and loyalty.

In the literature, Kozak defined the intention to recommend as when visitors were satisfied with the destination and were willing to visit again, and also as a personal preparation, or as a willingness to repeat the same destination again [37]. Hence, this study defines the intention to recommend as those satisfied with their participation in the arts and culture festivals, and their willingness to re-participate. As a result of the above discussion, the following hypothesis is presented:

**H3.** *Brand experience positively influences intention to recommend.*

### 3. Research Method

#### 3.1. Data Collection and Sampling Method

In order to examine the hypothesized relationship among the variables, a quantitative method using a survey questionnaire was applied. These were then distributed to a handiness sample of 201 individuals who had visited the Kaohsiung Cultural Center, The Pier-2 Art Center, and the Dadong Arts Center, for the 2019 Kaohsiung Spring Arts Festival (KSAF). Pier-2 Art Center, which was built in 1973 as communal port storerooms. They

were once abandoned and buried in history due to the city move from an industrial based to the service sector. Thus, the organizers re-discovered these Pier-2 storerooms just nearby the harbor. Later in 2001, a group of local artists recognized the Pier-2 Artistic Development Association, targeting this center to be the art special zone for artistic development in southern Taiwan.

In 2006, the Bureau of Cultural Affairs of Kaohsiung City Government took over and became the management entity. Ever since, The center positions itself as an international art platform, oriented towards avant-garde, experimental and original, open to creators from all over the world, making the area a place where tourists and locals enjoy art together. With the collision of an old area and new fine art, this center thus transforms itself into an art venue representative in Taiwan, an art area full of new vitality and liveliness.

Using the convenience sampling technique, all the respondents agreed to participate in the study. Of the initial 218 respondents who had participated in the KSAF, 17 were excluded as a result of missing ratings, leading to a final sample of 201 respondents who had completed both the initial and the follow up questionnaires. Kaohsiung was chosen because it hosts the largest arts and cultural festival in southern Taiwan, the KSAF held annually held by the Bureau of Cultural Affairs of the Kaohsiung City Government. The special performances combined arts, music, drama, dance, and traditional drama.

### 3.2. Measures

The authors proposed a conceptual model that drew upon previous research about the tourism brand identity and brand experience in arts and culture festivals. The questionnaire items were adapted based on the literature of the tourism brand identity, brand experience, and intention to recommend. The scale drew upon the aforementioned literature to propose a measure of arts and cultural festivals. The questionnaire was divided into two parts. The first part consisted of the scale of tourism destination brand identity, brand experience, and intention to recommend, and the second part collected demographic data. Before conducting the survey, pilot testing was conducted to ensure the validity and reliability. Confirmatory factor analysis (CFA) and structural equation modeling (SEM) were applied to examine how the tourism brand identity and the brand experiences affected the visitors' intention to recommend.

### 4. Research Results

### 4.1. Descriptive Analyses

Table 1 presents the demographic characteristics of respondents, including gender, age, frequency of attendance, income, and education. Female respondents (63.2%) outnumbered their male counterparts (36.8%). The majority of respondents were in the age group of 21–40 (57.7%). In education, 84% of the respondents had completed a Bachelor degree. Respondents making less than NT$20,000 a month comprised 52.2% of the sample, followed by NT$20,001–NT$30,000 (25.9%). For the majority of (57.7%), it was their first time attending these events. Of those respondents attending two or more times comprised 42.3% of the sample.

**Table 1.** Demographic characteristics of the respondents (N = 201).

|  | Frequency | Percent |
| --- | --- | --- |
| Gender |  |  |
| Men | 74 | 36.8 |
| Women | 127 | 63.2 |
| Age (years) |  |  |
| <20 | 51 | 30.3 |
| 21~40 | 116 | 57.7 |
| 41~60 | 21 | 10.4 |
| 61 and over | 3 | 1.5 |

**Table 1.** *Cont.*

|  | Frequency | Percent |
|---|---|---|
| Education |  |  |
| High school or less | 53 | 26.4 |
| Bachelor | 137 | 68.2 |
| Masters and above | 11 | 5.5 |
| Frequency of attendance |  |  |
| once | 116 | 57.7 |
| 2–3 times | 51 | 25.4 |
| More than 3 times | 34 | 16.9 |
| Income (NTD) |  |  |
| Less than 20,000 | 105 | 52.2 |
| 20,001–30,000 | 52 | 25.9 |
| 30,001–40,000 | 23 | 11.4 |
| More than 40,001 | 21 | 10.5 |
| Total | 201 | 100.00 |

*4.2. Correlation Analysis*

Table 2 shows the correlation coefficients of the observed variables. For the tourism brand identity, the measurement indicators (BI1~BI2) of the cross-correlation were 0.772, which was significant (t > 1.96), meaning that the two variables had the same potential factors. The correlation between the two variables was therefore positive.

**Table 2.** Mean, Standard Deviation and Correlation Analysis.

| Variables | BI1 | BI2 | EX1 | EX2 | EX3 | EX4 | Rev |
|---|---|---|---|---|---|---|---|
| BI1 | 10.00 |  |  |  |  |  |  |
| BI2 | 0.772 ** | 10.00 |  |  |  |  |  |
| EX1 | 0.581 ** | 0.635 ** | 10.00 |  |  |  |  |
| EX2 | 0.579 ** | 0.623 ** | 0.785 ** | 10.00 |  |  |  |
| EX3 | 0.540 ** | 0.615 ** | 0.657 ** | 0.653 ** | 10.00 |  |  |
| EX4 | 0.553 ** | 0.618 ** | 0.618 ** | 0.745 ** | 0.745 ** | 10.00 |  |
| Rec | 0.650 ** | 0.616 ** | 0.647 ** | 0.614 ** | 0.574 ** | 0.555 ** | 10.00 |
| Mean | 30.595 | 30.488 | 30.950 | 30.889 | 30.739 | 30.678 | 30.829 |
| SD | 0.681 | 0.685 | 0.607 | 0.646 | 0.651 | 0.629 | 0.681 |
| Skewness | 0.045 | −0.049 | −0.243 | −0.145 | −0.112 | −0.163 | −0.010 |
| Kurtosis | −0.197 | 0.137 | 0.127 | −0.380 | −0.326 | 0.043 | −0.529 |

Note: BI1 and BI2 measure tourism destination brand identity. EX1–EX4 measure brand experience. Rec measures tourists' behavior. ** $p < 0.01$.

The cross correlation of the brand experience (EX1~EX4) was between 0.618~0.785, which was significant (t > 1.96), and indicated that the four observed variables had the same potential factors. The cross correlation of intention to recommend was 0.813, which was significant (t > 1.96), and showed that the variables both had the same potential factors.

*4.3. Model Fit Indicators of the Confirmatory Factor Analysis*

According to the literature, three types of indicators should be used for evaluating goodness-of-fit [42,43]. The model fit evaluation was based upon the procedure described in Byrne [44]. Absolute fit measurement indices are presented in Table 3. The $\chi^2$ goodness-of-fit test evaluates the adequacy of the theorized model creation of a covariance matrix and estimated the coefficients when compared to the observed covariance matrix. However, since the sample size may influence the value of $\chi^2$, a larger sample size can make the test

deficient for adequately assessing the model fitness [42]. Address to fix this propensity involved the procedure of dividing the value of $\chi^2$ by the degrees of freedom (DF) [45,46]. Bagozzi and Yi suggested that any $\chi^2$/DF evaluation of less than 5 was tolerable for a large sample [45]. The goodness-of-fit calculation of ($\chi^2$/DF = 4.46) for this prototypical model specified a positive result. With two further measures connected to the AGFI and RMSEA, the values for the AGFI = 0.817 fall within the range of acceptable values, but close to 0.90, and the values for the RMSEA = 0.132 should be within the tolerable range. Therefore, the primarily hypothesized model (see Figure 1) fits the data well.

**Table 3.** Absolute fit measures.

|  | Criteria | Indices |
|---|---|---|
| 1. CMIN/DF | 1~5 | 4.46 |
| 2. GFI | ≥0.90 | 0.913 |
| 3. AGFI | ≥0.90 | 0.817 |
| 4. RMR | <0.08 | 0.016 |
| 5. RMSEA | ≤0.10 | 0.132 |

Note: RMSEA ≤ 0.05 representative of excellent fit; 0.05~0.08 representative of good fit; 0.08~0.10 indicates moderate fit; >0.10 represents poor fit.

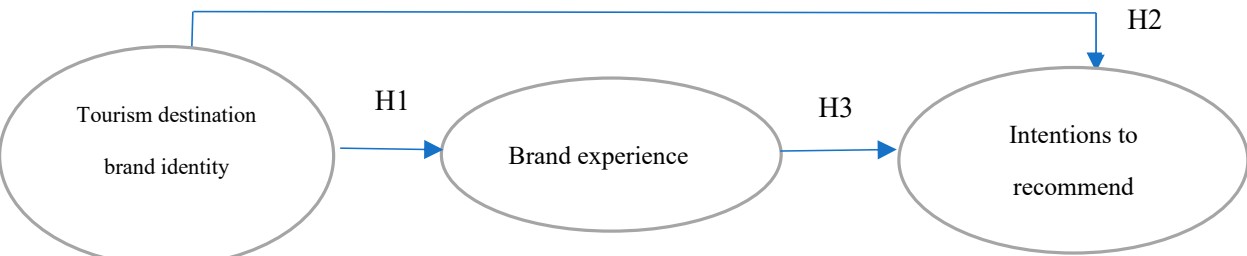

**Figure 1.** The conceptual model in the present study.

Relative fit measurement indices are presented in Table 4. For the NFI, CFI, and IFI, RFI ≥ 0.90 were originally considered representative of a good-fitting model. The values in the AMOS output (NFI = 0.939, CFI = 0.951 and IFI = 0.952) indicated that the model fit the data well. With one further measure related to the RFI, the value for RFI = 0.899 falls within the range of acceptable values but was close to 0.90, and was thus in the acceptable range. Therefore, the initially hypothesized model fits the data well. Although it has not reached the acceptable value of 0.08, it has not yet exceeded 0.15, the fit measurement of RMSEA value showing that the mode is still acceptable.

**Table 4.** Relative fit measures.

|  | Criteria | Indices |
|---|---|---|
| 1. NFI | ≥0.90 | 0.939 |
| 2. CFI | ≥0.90 | 0.951 |
| 3. IFI | ≥0.90 | 0.952 |
| 4. RFI | ≥0.90 | 0.899 |

Table 5 presents the parsimonious fit measures PNFI and PGFI, which were >0.5, and were considered representative of a good model fit.

Overall, the three types of measures show that this model has a good fit with sufficient construct validity.

**Table 5.** Parsimonious fit measures.

|  | Criteria | Indices |
|---|---|---|
| 1. PNFI | ≥0.50 | 0.570 |
| 2. PGFI | ≥0.50 | 0.578 |

### 4.4. Structural Model Results

The overall structural model was tested using AMOS 18.0 on the 25-item model. The reliability and factor analysis was used to develop a model (Figure 2), which proposed two components of tourism destination brand identity: (1) brand perception, and (2) brand self-concept. These were directly related to the overall value of brand experience and behavior of the visitor. In this study, there were four proposed brand experience aspects: (1) sensory experience, (2) affective experience, (3) intellectual experience and (4) behavioral experience, which have direct effects on the intention to recommend. The indirect effects link the tourism destination brand identity components with the extent to which the brand perception and brand self-concept stimulated interest in the arts and culture, and affected the intention to recommend.

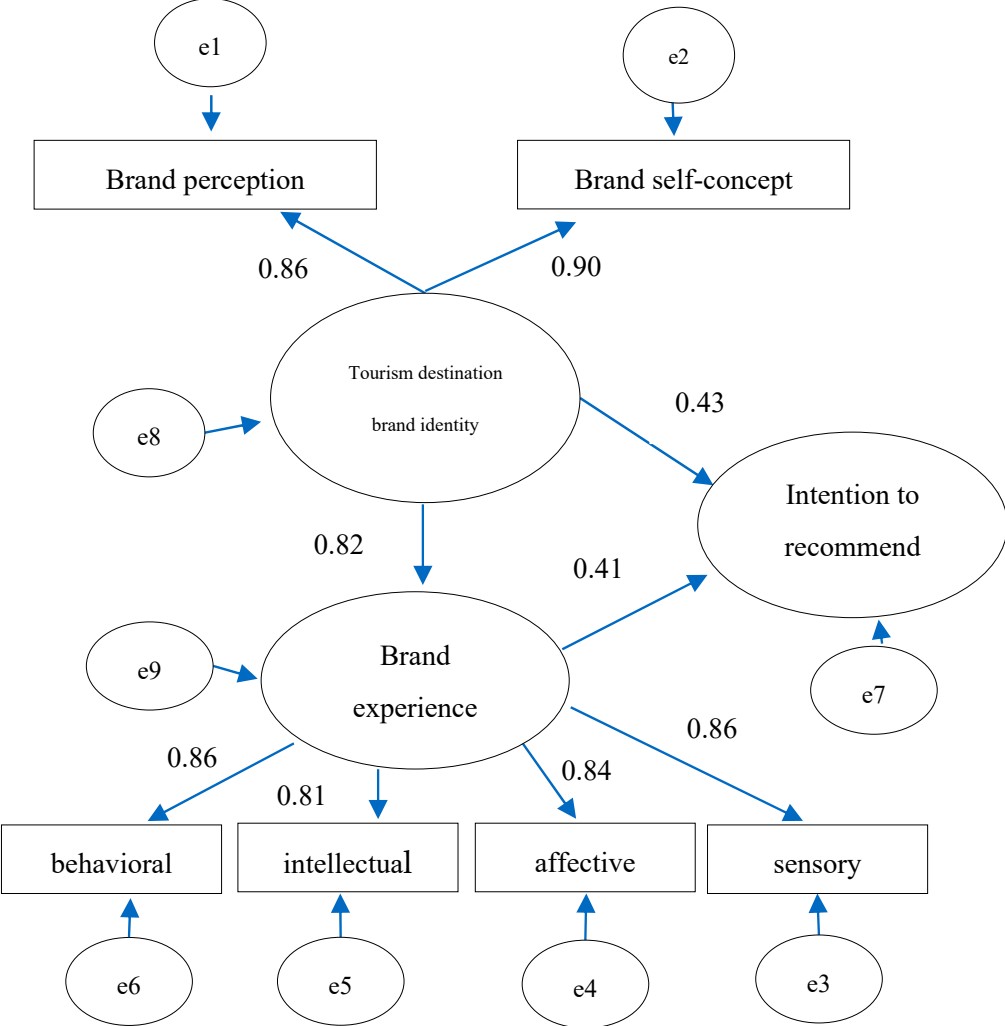

**Figure 2.** Structural equation model.

The key objectives of the present research are to explore the links between destination brand identity, brand experience and behavior of cultural visitors at a cultural and arts tourism destination. Given the results of structural model in the research, it may be assumed that a lot still needs to be done for the creating (1) sensory experience, (2) affective

experience, (3) intellectual experience and (4) behavioral experience, so that their overall value of brand experience can contribute to the intention to recommend of visitors that may lead to better development at a cultural and arts tourism destination.

There had been several applications of the SEM to tourism in varying contexts over recent years, confirming the utility and effectiveness of the method [47–49]. In particular, brand perception, and brand self-concept of destination brand identity was recognized as an influential part in relating to the overall art and cultural experience.

The calculation of the internal consistency reliabilities for the destination brand identity items provides an indication of great item consistency. This analysis resulted in coefficients (Alpha = 0.927) for all scales of destination brand identity. The total scale reliability of brand experience was reported as a coefficients (Alpha = 0.909), which is considered very good in a statistical research. The results show that the coefficients (Alpha = 0.925) for intention to recommend items.

The dimension of the tourism destination brand identity includes two aspects, brand perception and brand self-concept. The estimate of brand perception was 0.86, with $R^2$ being 0.739, and the estimate, higher than 0.5, had a high explanatory ability. The estimate of brand self-concept was 0.9, with $R^2$ being 0.81, and the estimate, higher than 0.5, had a high explanatory ability. Further, a t value larger than 1.96 and was therefore significant. One can see from the estimate of each aspect that the brand self-concept (0.9) was the most important factor in the tourism brand identity, followed by the brand perception. This result showed that it is an important factor to assess the tourists' perceptions and attitudes toward arts and culture festivals, own-brand marketing strategies, along with its images, symbols, patterns, and characters. If the organizers sought to increase their brand identity, they needed to take the festival brand into account, as argued by Blain, Levy, and Ritchie [12]. The path coefficient from the tourism brand identity towards the brand experience was 0.82 (standardized coefficient = 0.82, t = 11.48 > 1.96).

Therefore, hypothesis 1 was supported, as the tourism brand identity appeared to have a direct and positive effect on brand experience. The estimate of intention to recommend was 0.95, with the $R^2$ being 0.90, and the estimate higher than 0.5, had a high explanatory ability, and a t value larger than 1.96 was significant. The relationship model (Figure 1) of tourism destination brand identity, brand experience, and behavioral intention can verify the hypotheses in this research. The path coefficient of tourism brand identity toward intention to recommend was 0.43 (standardized coefficient = 0.43, t = 3.37 > 1.96). Hence, hypothesis 2 was supported. Brand identity appears to affect the intention to recommend. The three hypotheses in this research are therefore supported.

The dimension of brand experience includes four aspects, sensory, affective, intellectual, and behavioral. The estimate of sensory, 0.86 ($R^2$ is 0.739), was higher than 0.5, and had a high explanatory ability. The estimate of affective, 0.84 ($R^2$ is 0.706), was higher than 0.5, and had a high explanatory ability. The estimate of intellectual, 0.81 ($R^2$ is 0.66), was higher than 0.5, and had a high explanatory ability. The estimate of behavioral, 0.76 ($R^2$ is 0.578), was higher than 0.5, and had a high explanatory ability. Additionally, the t value estimate of the affective, behavioral, and intellectual was larger than 1.96 and was significant. One can learn from the estimate of each aspect that the sensory aspect (0.86) was the crucial factor in the tourists' cognition towards the brand experience, followed by the affective experience (0.84), intellectual (0.81), and behavioral experience (0.76). This result shows that it was most important to rate the tourists' sensory experience of the arts and culture festival, followed by the other aspects. The path coefficient for the brand experience towards the intention to recommend was 0.41 (standardized coefficient = 0.41, t = 3.42 > 1.96).

Therefore, hypothesis 3 was supported. In point of fact, the better the brand experience of those who participate in arts and culture festivals, the higher their willingness to visit the destination.

## 5. Conclusions

This study aimed to determine the key aspects of tourism destination brand identity, and brand experience in the tourists' intention. Based on the outcome, it is important to assess their perceptions and attitudes toward arts and culture festivals, own-brand marketing strategies, along with its images, symbols, patterns, and characters. These findings were consistent with Blain, Levy, and Ritchie who contended that brand perception was the key to developing location tourism and the marketing core [12]. Moreover, if tourists were more satisfied with the brand experience, their intention to recommend will, indeed, increase. Based on these findings, it is concluded that the type of tourism destination brand identity, are anticipated in organizing activities, events and festivals at the destination level to help in determining visitors' overall brand experience. Results of the present study indicate that visitors may have greater satisfactory experience based on a perceived positive tourism destination brand identity.

Among all of the factors, brand experience seems to have a greater and more positive effect on the tourists' intention to recommend. Tourists agree that the destination can create an unforgettable experience destination, and it can also enhance the emotional connection between them and the art and festival destination, as they were more than willing to participate in the related activities in the future. The motivation for those participating in these festivals enhances the festival's reputation, along with experiencing the beauty of art and learning new facts and interesting information. These findings are consistent with Bigné et al., Grappi and Montanari and Prayag and Ryan who proposed that recommending was the positive result of a satisfactory travel experience [47,49,50]. This has proven that tourists have had a positive attitude towards the arts and culture festivals, and that brand identity and brand experience positively affected the tourists' behavioral intentions. This suggests that when a tourism destination is positive recognized by visitors, their evaluation of travel experience as becomes notable aware, visitors are able to make more accurate predictions about its tourism brand value based on their actual experiences. The study revealed that by enhancing the tourism destination brand identity, the tourists will generate a positive evaluation of the destination, and be willing to recommend, as it will also enhance their loyalty towards the destination.

This study has made some significant contributions to the emerging body of research in arts and culture festival and brand management. The current study empirically tested the role of the tourism destination brand identity on building brand loyalty, of which the number of studies in this area is minimal. Moreover, the study found that brand perception was the most important factor in tourism destination brand identity, followed by brand perception, which can result in a greater brand experience and behavioral intention. In terms of the practical applications, especially through the process of tourists' participation, this can provide a deeper understanding of cultural awareness and recognition. It is also recommended that there should be increased brand perception and brand self-concept connected activities for the cultural and arts events in the form of marketing and promotion of their products or experience.

Through the market mechanism, industrial mechanism, the applied brands, the visitors experience to activate the arts and cultural tourism, and resources, is an important task. One of the strategies that can be used to effectively increase the tourism destination brand identity may be through increased and open communication activities at social media sites and platforms. This may imply that this cultural awareness of many cultural and arts events is considered to more likely to refer to distinct components of local cities and cultural and arts sustainability. It may be possible that this can make a contribution to the creative industries, arts and culture festivals and the local economic development. In 2020, It is a critical issue generated from the COVID-19 pandemic for regions and destinations marketers to provide high-quality tourist experiences to reduce the perceptions of travel risks for art and cultural travelers. In this viewpoint, the study outcomes have significant management and marketing suggestions for tourism authorities, which have the chance to make recommendations to the many tourism areas in Taiwan. Further, the present

study has not focused on Taiwanese art and cultural tourists spent time more on overall art and cultural events at a tourism destination and further related studies may be need to investigate travel risks for art and cultural travelers before their decides to participate various tourism and cultural events.

Consequently, exploring statistical information might lead to a primary explanatory understanding of the relationship between categories of brand experience and behavioral intention. This study will be useful for managers in the arts and cultural tourism sector, whilst also providing insights to stimulate further research.

Even though every effort has been made to make the current research as inclusive as possible, there are some limitations which should be considered. There is also limitation of the present research, a critical question remains—would greater brand experience formed in their perception of service evaluation at a cultural and art tourism destination? It is probable that their perception of service evaluation drivers of different brand experience and behavioral intention that may need to be investigated by more future studies. The results of this study may not be generalizable to other festivals in other communities, comparative studies between period of time and the small number of respondents. Furthermore, the measurement instrument was a questionnaire. Consequently, follow-up researchers may well consider future research directions and different control variables.

**Author Contributions:** Conceptualization, C.-T.C. and Y.-C.C.; methodology, C.-T.C. and Y.-C.C.; software, C.-T.C. and Y.-C.C.; validation, C.-T.C. and Y.-C.C.; formal analysis, C.-T.C. and Y.-C.C.; investigation C.-T.C. and Y.-C.C.; resources, C.-T.C. and Y.-C.C.; data curation, C.-T.C. and Y.-C.C.; writing—original draft preparation, C.-T.C. and Y.-C.C.; writing—review and editing, C.-T.C. and Y.-C.C.; visualization, C.-T.C. and Y.-C.C.; supervision, C.-T.C. and Y.-C.C.; project administration, C.-T.C. and Y.-C.C.; funding acquisition, C.-T.C. and Y.-C.C. All authors have read and agreed to the published version of the manuscript.

**Funding:** This research received no external funding.

**Institutional Review Board Statement:** Not applicable.

**Informed Consent Statement:** Not applicable.

**Data Availability Statement:** Not applicable.

**Conflicts of Interest:** The authors declare no conflict of interest.

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
