# Peer review of "The Effect of Destination Brand Identity on Tourism Experience: The Case of the Pier-2 Art Center in Taiwan"

_sustainability, doi:10.3390/su15043254_

Round 1

Reviewer 1 Report

Dear Authors of the paper entitled "The Effect of Place Brand Identity on Tourism Experience: The Case of The Pier-2 Art Center in Taiwan".

Thank you for the opportunity to review your research proposal.

After reading your manuscript and after an initial documentation for this review, I have the following concerns:

First of all, there are some similarities between your document and the article published at the address: https://doi.org/10.1016/j.jdmm.2018.06.004.

I recommend you to revise the sequences between rows 108-109, 201-204, 236-239 and try to decrease the degree of similarity.

One major issue of your manuscript consists in the outdateness of the references from the previous literature. Thus, you don't have any resource from the years 2022 and 2021, just one from 2020, and the rest of the references are older than 2019.

In order to improve the context of your research proposal, I recommend you to include the following references: https://doi.org/10.3390/joitmc8010028, https://doi.org/10.3390/joitmc8010052, https://ideas.repec.org/a/ddj/fseeai/y2014i1p39-46.html, https://doi.org/10.3390/land10121319.

At rows 84-86 you say that "According to recent survey findings, communicate with the destination via an unforgettable travel experience and enhance the emotional connection between the tourist and the destination of the festival (Blain, Levy, Ritchie, 2005)."

Please revise this sentences because you start with "...recent findings...", but at the end of the sentence you have a reference from 2015. This is not "recent".

The Introduction should contain the description of the research goal and the research question. I tried to identify them, but they are not clearly described. Please revise this issue.

The title of the sub-section 2.1 "place brand identity of art and creative tourism" should start with capital letter.

Regarding the constructs of the model and the survey, I recommend to add an appendix at the end of the article where you present the questionnaire, so that the readers know the questions.

The Conclusion section should also contain the future research directions, based on your research results.

The limitations should also include the small number of respondents.

Best Regards!

Author Response

First of all, there are some similarities between your document and the article published at the address: https://doi.org/10.1016/j.jdmm.2018.06.004.

I recommend you to revise the sequences between rows 108-109, 201-204, 236-239 and try to decrease the degree of similarity.(Yes,has been revised that)

One major issue of your manuscript consists in the outdateness of the references from the previous literature. Thus, you don't have any resource from the years 2022 and 2021, just one from 2020, and the rest of the references are older than 2019.

In order to improve the context of your research proposal, I recommend you to include the following references: https://doi.org/10.3390/joitmc8010028, https://doi.org/10.3390/joitmc8010052, https://ideas.repec.org/a/ddj/fseeai/y2014i1p39-46.html, https://doi.org/10.3390/land10121319.

(Have include the references)

At rows 84-86 you say that "According to recent survey findings, communicate with the destination via an unforgettable travel experience and enhance the emotional connection between the tourist and the destination of the festival (Blain, Levy, Ritchie, 2005)."

Please revise this sentences because you start with "...recent findings...", but at the end of the sentence you have a reference from 2015. This is not "recent".(deleted "recent")

The Introduction should contain the description of the research goal and the research question. I tried to identify them, but they are not clearly described. Please revise this issue.(have been revised that)

The title of the sub-section 2.1 "place brand identity of art and creative tourism" should start with capital letter. (Changed that,thanks)

Regarding the constructs of the model and the survey, I recommend to add an appendix at the end of the article where you present the questionnaire, so that the readers know the questions.

The Conclusion section should also contain the future research directions, based on your research results.

The limitations should also include the small number of respondents.

(Revised that)

Reviewer 2 Report

The topic of the article is interesting, but the text has a number of limitations. The abstract contains rather vague information, we recommend a clearer synthesis of the purpose, usefulness and results/originality of the research. The statistical evidence in the introduction is from 2019 or earlier, so before the period of the SARS COV2 pandemic. As the article is intended to be published in early 2023, the timeliness of the article with respect to a situation in 2019 should be justified by a reference to the impact of the pandemic on tourism in Taiwan in 2020-2022. Incidentally, the entire bibliography is before 2020, except for 2 titles from 2020. The two titles are referenced in the text, but, surprisingly, they do not appear in the final bibliography (Escobar, Lopez, Manuel, & Garcia, 2020, Zach and McGehee, 2020, respectively). Authors should clarify this and supplement the bibliographic references in the literature review and other relevant parts of the paper with bibliographic titles newer than 2019.

In line 96, the word recommend appears twice, and the phrase must be reformulated. Same with lines 208-210. In the methodology part, it should be explained more clearly how the questionnaire items were constructed, which the authors do not present in the paper. In table 1 there is a sign that is not in English, probably meaning "total". In Table 3, RMSEA should be less than 0.10, and it is 0.132. The text says it's within acceptable limits. You have to scientifically justify the fit of the model. Lines 320-324 apply the importance of using SEM in tourism, without making concrete reference to the analyzed model. The conclusions must be formulated much more concretely, clearly specifying the contributions of the study, the novelty brought to the updated literature 2022. Lines 397-398 say that previous studies are few, should be presented. The limitations of the study (the sample size would be one) are not sufficiently presented, nor are the clear contributions to the development of the field.

Author Response

Dear Sir

According to your comments have been revised in red line please see the attachment

Reviewer 3 Report

Thank you for the opportunity to review the article “The Effect of Place Brand Identity on Tourism Experience: The Case of The Pier-2 Art Center in Taiwan”. The paper addresses an interesting and well researched theme in the recent period about the implications of brand identity and brand experiences which can influence visitors’ intention to recommend and to support the tourism sector or to enhance the attractiveness of cultural and creative sector. The paper is also in line with the section “Tourism, Culture, and Heritage” and with the special issue was submitted to: “Culture, Tourism and Leisure Behavior”.

This study represents a new approach in the field, discussing the subject that needs to be comprehensively analyzed because “brand identity has been widely discussed and applied”, as the authors underline.

Also, the study is written in an adequate manner and the results are presented clearly and coherently, using visuals and text. The figures and tables presented in the paper were relevant to explore the results of the research and the ways these were adapted to the explanations in the text.

As the authors say, one conclusion assumes that “visitors may have greater satisfactory experience based on a perceived positive tourism place brand identity” but also inform the readers about limitations of the study.

Moreover, there are some important observations that should be addressed in this revision.

-          Please revise and correct all the entries of the cited works and References section to the publication standards and style.

-          Some of the cited materials are not presented in the References section: e.g., World Travel & Tourism Council (2019) or are presented wrong: e.g., Allen, O'Toole, Harris and McDonnell (2011) for the incomplete reference: Allen, J., Harris, R., McDonnell, I., & O’Toole, W. (2011). Event Management; or cited Country Brand Index, 2009 and the reference is 22. Country Brand Index (2006).

-          The introduction is outdated using the 2019 statistics from World Travel & Tourism Council. Please find 2021 or 2022 references

-          The main issue of the paper is the wrong understanding and the usage of different terms: “place branding” and “destination branding” (see S. Hanna & J. Rowley, 2008, p. 64; G. Hankinson, 2005; G. Kerr, 2006, p. 277; Briciu, 2013, p. 9; S.-Y. Park & J. F. Petrick, 2006; H. Qu et al., 2011, p. 466). In this paper both terms are seen as synonyms: e.g., Line 164-170 “Accordingly, within Cai’s proposed with destination branding the marketing function’s role is emphasized, in a tourism destination’s brand identity progress special consideration should be given to examining the specific characteristics of the brand as an organization, which should statement the topic of destination culture, its regional people (Anholt 2002; Konecnik 2004) and their relationship. Hwang and Lyu (2015), also indicated that the tourism place brand identity positively affects the willingness to recommend a destination.” I think that the philosophy of the research gravitates around tourism and destination, so starting from the Title, Abstract, Keywords and the rest of the paper the term place should be modified with destination. Both terms should be clearly defined in the paper and emphasize the role of the destination as other authors already showed us these differences. More in-depth differences are about “destination branding” and “location branding” as more than “place marketing”, “destination marketing”, “locational marketing”. Even more clarifications can be found in the specific literature about variations of place branding as “thematic branding”, “regional branding”, “geo-branding”.

-          Future directions of the research should be better described and also to highlight the novelty factor better.

-          After reading the title about the “The Case of The Pier-2 Art Center in Taiwan”, I found no details in the paper about that subject, the objectives are not direct related to this subject, the discussion and the conclusions are not presenting more in-depths results about The Pier-2 Art Center but are focusing more about creative industries, arts and culture festivals.

Author Response

(The authors gave the same response as above.)

Reviewer 4 Report

The topic of this research is related to the themes of this journal. Please find enclosed my constructive remarks for your guidance.

I noticed that the researchers provided a short background. They could have included more recent references, including from this journal within the introduction and in other areas.

The introductions should better clarify the research objectives. What is the rationale of this study? Are there any studies on this topic? The authors are expected to elaborate further on the contribution of this research.

The literature review can include the following references:

Camilleri, M. A. (Ed.). (2019). Tourism planning and destination marketing. Bingley: Emerald Publishing, UK.

Camilleri, M. A. (Ed.). (2019). The branding of tourist destinations: theoretical and empirical insights. Bingley: Emerald Publishing, UK. 

The hypotheses could have been presented separately. The authors should have featured at least 2-3 paragraphs before formulating each hypothesis.

The authors could have indicated the source of their measures. It could have been better if they presented them as well.  Moreover, they did not clarify the results of the reliability (Alpha and CR) as well as validity (AVE) and discriminant validity of the constructs. Where did they disclose these figures?

The conclusions should have presented at least two separate areas. The authors could have discussed on the theoretical and practical implications. What about the limitations and future research directions?

Author Response

(The authors gave the same response as above.)

Round 2

Reviewer 1 Report

Dear Authors,

After reading and analyzing the revised version of your manuscript, I have the following recommendations that you should carefully address:

1. There are some similarities between your text and some already published articles.

For example, in your manuscript:

- the rows 34 - 47, 89 -95, 101 - 111, 200 - 202, are similar to https://doi.org/10.1016/j.jdmm.2018.06.004.

- the rows 255 - 277 are similar to existing presentation from the address: https://artres.moc.gov.tw/en/database/twContent/f86d7d55585747d5a1075a04e7a92f1d?pageLang=en

- the rows 324 - 331 are similar to https://doi.org/10.1016/j.tourman.2013.03.008.

I warmly recommend you to revise these sequences of text from your manuscript and to decrease the degree of similarity.

2. As I already told you in the previous round of review, I recommend you to include in your article the following relevant resources: https://doi.org/10.3390/joitmc8010052, https://ideas.repec.org/a/ddj/fseeai/y2014i1p39-46.html, https://doi.org/10.3390/land10121319. By including these resources, you will add valuable references to your article and the readers will understand in a better manner the context of your research.

3. At the row 119 you say that "As a result of the above discussion, the following hypothesis is presented:". And then, you present 3 hypotheses. Please correct the initial sentence to "As a result of the above discussion, the following hypotheses are presented:", because you have many hypotheses, not just one.

Dear Authors,

I recommend you to address all the above mentioned recommendations so that your manuscript becomes a scientific article.

Best Regards!

Author Response

Dear reviewers

We have revised the article according to your comments, please see the attachment.

Best regards,

Reviewer 2 Report

Dear authors, thank you for the reply letter, but a number of observations have not been modified. In our opinion, a paragraph dedicated to the effect of the COVID 19 pandemic on tourism in this destination would be useful. Perhaps the experience of the cultural tourist in Pier-2 Art Center was not significantly influenced by the COVID pandemic, but for an article that wants to be published in 2023, the period 2020-2022 cannot be omitted. In lines 31-32, you've updated the statistic, but in line 33 you've kept a claim supported by a 2018 statistic. An answer is needed regarding the addition of only 2 titles after 2020 and a statistical justification for RMSEA=0.10.

Author Response

(The authors gave the same response as above.)

Reviewer 3 Report

Thank you for the opportunity to review the revised version of the paper “The Effect of Place Brand Identity on Tourism Experience: The Case of The Pier-2 Art Center in Taiwan”.   

The authors responded and explained with reasonable arguments and corrected all the remarks and observations highlighted in the previous review and the results suggest a more consistent and logical text.

To sum it up, the authors developed a more in-depth theoretical presentation about the subject, integrating the suggested aspects of the review.

I consider that the paper is publishable after a final check from the authors, the English add-ons are not quite grammatically correct. Some of the references must be corrected by using the standards and guidelines of the journal.

Author Response

(The authors gave the same response as above.)

Reviewer 4 Report

I noticed that you have refined your contribution in many areas. Well done and congratulations. I am recommending the acceptance of this manuscript.

Author Response

Thanks for your encouragement 

Round 3

Reviewer 1 Report

Dear Author(s),

The paper still has some issues, according to the following remarks:

1. There are some similarities between the texts of your manuscript and the article published at the address: https://doi.org/10.1016/j.jdmm.2018.06.004. For example, the sequences from your manuscript between the rows 90 - 94, 111 - 116, 220 - 226.

The rows 276 - 281, are very similar to https://artres.moc.gov.tw/en/database/twContent/f86d7d55585747d5a1075a04e7a92f1d?pageLang=en.

I recommend you to revise these paragraphs and to decrease the degree of similarity.

2. The following references from your final bibliography must be completed/corrected, according to the citation standards of the Sustainability Journal: reference 28 (pages of the article), reference 31 (the title of the article, the initial of the author's first name and the web reference), reference 79 (pages of the article).

3. In the Conclusion section, the future research directions should be better exposed. I recommend you to think about comparative studies between period of times. Please also think about different control variables.

Best Regards!

Author Response

Dear reviewers

we have revised the paper according to your comments, please see the attachment

Best regards

Reviewer 2 Report

The authors have modified and added aspects noted in the previous review. I recommend publication in this revised form.

Author Response

Thank you very much